A new nematode species, Chromadorina tangaroa sp. nov. (Chromadorida: Chromadoridae) from the hull of a research vessel, New Zealand

http://orcid.org/0000-0001-5310-9198 Leduc Daniel Daniel.Leduc@niwa.co.nz
National Institute of Water and Atmospheric Research , Wellington , New Zealand
Costello Mark
Electronic publication date: 2020 May 26
Publication date: 2020
Volume: 8
Electronic Location ID: e9233
Received 2020 Mar 27; Accepted 2020 May 2
Copyright: © 2020 Leduc
Copyright year: 2020
Copyright holder: Leduc
License: This is an open access article distributed under the terms of the Creative Commons Attribution License, which permits unrestricted use, distribution, reproduction and adaptation in any medium and for any purpose provided that it is properly attributed. For attribution, the original author(s), title, publication source (PeerJ) and either DOI or URL of the article must be cited.
License URL: https://creativecommons.org/licenses/by/4.0/

Keywords: Biofouling, New Zealand, New species, Taxonomy, Nematoda, Epiphytic

Funding: NIWA’s Coasts and Oceans Centre Research Programme Funding was provided by NIWA’s Coasts and Oceans Centre Research Programme ‘Marine Biological Resources’. The funders had no role in study design, data collection and analysis, decision to publish, or preparation of the manuscript.

==============================
Chromadorina is a globally distributed, largely marine nematode genus frequently found on a variety of organisms, including macro- and microalgae and crustaceans, as well as artificial substrates such as settlement plates and ship hulls. Here, Chromadorina tangaroa sp. nov. is described from filamentous seaweed growing on the hull of RV Tangaroa anchored in Wellington, North Island of New Zealand. It is characterized by body length 763–1,086 μm, and pore of secretory-excretory system located at or near level of teeth. Males have spicules with rounded capitulum followed by a narrower shaft and blade tapered distally, a gubernaculum as long as the spicules, and three cup-shaped precloacal supplements, and females are characterized by a cuticularized prevulvar pad, vagina located at 46–48% of body length from anterior, and vagina anteriorly directed. Chromadorina tangaroa sp. nov. is the first species of the genus to be described from New Zealand, but it is unclear whether it is native to the region because it may have dispersed as part of ship hull biofouling communities. Long-distance transport of nematodes through ship hull biofouling may be a common occurrence, but too little is known about the occurrence of nematodes on ship hulls to gauge the potential effect of shipping on nematode species distributions.

Introduction

Vessel hulls are colonized by a wide variety of sessile and mobile organisms ranging from microscopic prokaryotes and unicellular eukaryotes to large invertebrates and macroalgae. These biofouling communities (i.e., accumulations of organisms on submerged artificial surfaces) are sometimes transported over large distances, which can lead to the introduction of non-indigenous organisms in new environments where they may establish new populations and potentially impact local ecological communities (Hewitt, Gollasch & Minchin, 2009; Gallardo et al., 2016). Most of the literature on biofouling has so far focused on relatively large, macrofaunal-sized organisms, as well as microorganisms (Dang & Lovell, 2000; Zardus et al., 2008; Farrapeira, Tenorio & Do Amaral, 2011). Limited information is available on meiofaunal organisms such as nematodes (Fonseca-Genevois et al., 2006; Chan, MacIsaac & Bailey, 2016; Von Ammon et al., 2018), even though they are common epibionts and arguably the most numerous animals in shallow sedimentary environments worldwide (Giere, 1993). Nematodes have very limited active dispersal abilities, but their small size allows them to be passively transported by currents into the water column (Shanks & Walters, 1997; Boeckner, Sharma & Proctor, 2009), or as epibionts on sea turtle carapaces and drifting macroalgae (Arroyo, Aarnio & Bonsdorff, 2006; Correa et al., 2014). These transport pathways are thought to largely explain the cosmopolitan distribution of some nematode and other meiofaunal taxa (the so-called “meiofauna paradox”; Giere, 1993). Long-distance transport of nematodes is also likely to occur through both biofouling and ballast slurry sediments (Radziejewska, Gruska & Rokicka-Praxmajer, 2006; Sutherland & Levings, 2013), but little is known about the occurrence of nematodes on ship hulls and the potential effect of shipping on nematode species distributions.

Hard surfaces such as rocks and artificial structures do not provide adequate substrates for nematode colonization (Heip, Vincx & Vranken, 1985); once they are colonized by biofilm-forming microorganisms and/or habitat-forming macroalgae and invertebrates, however, they constitute a suitable substrate for a range of epibiotic nematodes (Jensen, 1984; Kito & Nakamura, 2001; Fonseca-Genevois et al., 2006; Majdi et al., 2011). Species of the nematode genus Chromadorina Filipjev, 1918 are found living on a variety of other organisms, including macroalgae (Chromadorina laeta (De Man, 1876) Micoletzky, 1924, C. obtusa Filipjev, 1918, C. epidemos Hopper & Meyers, 1967) (Filipjev, 1918; Hopper & Meyers, 1967), vascular plants (C. epidemos Hopper & Meyers, 1967; C. erythrophthalma (Schneider, 1906) Wieser, 1954) (Hopper & Meyers, 1967; Jensen, 1979, 1984), periphyton (C. hiromii Kito & Nakamura, 2001, C. viridis (Linstow, 1876) Wieser, 1954, C. bioculata (Schultze in Carus, 1857) Wieser, 1954) (Andrássy, 1962; Kito & Nakamura, 2001), and the gill cavity of spider crabs (C. majae Wieser, 1968) and crayfish (C. astacicola (Schneider, 1932) Wieser, 1954) (Wieser, 1968; Schneider, 1932). Chromadorina has also been recorded on artificial settlement plates (Fonseca-Genevois et al., 2006; Von Ammon et al., 2018) and ship hulls (Chan, MacIsaac & Bailey, 2016). Most species are marine, except for a few which are found in lakes (C. astacicola), groundwater (C. bercziki) or both freshwater and brackish waters (C. bioculata and C. viridis). Here, we describe a new Chromadorina species recovered from filamentous seaweed growing on the hull of RV Tangaroa anchored in Wellington, North Island of New Zealand.

Materials and Methods

Samples of macroalgae were collected by divers from the hull of RV Tangaroa while anchored at Burnham Wharf in Wellington Harbour on 10 November 2019. The macroalgal material containing nematodes was fixed in 10% buffered formalin. Nematodes were manually sorted from the algal material using a dissecting microscope (×50 magnification) and transferred to glycerol and mounted onto permanent slides (Somerfield & Warwick, 1996).

Measurements were obtained using an Olympus BX53 compound microscope with cellSens Standard software. All measurements are in μm, and all curved structures are measured along the arc. The terminology used for describing the arrangement of morphological features such as setae follows Coomans (1979). Type specimens are held in the NIWA Invertebrate Collection (Wellington), and the National Nematode Collection of New Zealand (Auckland).

The electronic version of this article in Portable Document Format (PDF) will represent a published work according to the International Commission on Zoological Nomenclature (ICZN), and hence the new names contained in the electronic version are effectively published under that Code from the electronic edition alone. This published work and the nomenclatural acts it contains have been registered in ZooBank, the online registration system for the ICZN. The ZooBank Life Science Identifiers (LSIDs) can be resolved and the associated information viewed through any standard web browser by appending the LSID to the prefix http://zoobank.org/. The LSID for this publication is: urn:lsid:zoobank.org:pub:5DC49B25-C878-4FBA-A74A-7BE2F61E31AB. The online version of this work is archived and available from the following digital repositories: PeerJ, PubMed Central and CLOCKSS.

Systematics

Order Chromadorida Chitwood, 1933

Family Chromadoridae Filipjev, 1917

Subfamily Chromadorinae Filipjev, 1917

Genus Chromadorina Filipjev, 1918

Generic diagnosis: (Modified from Tchesunov (2014)) Homogeneously punctated cuticle with transverse rows of punctations, no lateral differentiation. Amphideal fovea when visible located near level of cephalic setae, transverse slit-like, unispiral, spiral, cryptocircular or loop-shaped. Buccal cavity with three equal teeth or with dorsal tooth slightly larger than ventrosublateral teeth. Ocelli and cup-shaped precloacal supplements may be present. Tail conical or conico-cylindrical with conspicuous spinneret.

Type species: Chromadorina obtusa Filipjev, 1918

Remarks: Previous diagnoses state that when visible, the amphideal fovea is slit-like (Platt & Warwick, 1988; Tchesunov, 2014). However, several species, including C. erythrophthalma (Schneider, 1906) Wieser, 1954, C. salina Belogurov, 1978, C. supralittoralis Lorenzen, 1969, and C. tangaroa sp. nov., have a unispiral, spiral cryptocircular, or loop-shaped amphideal fovea. Venekey et al. (2019) provided a list of 27 valid Chromadorina species.

Chromadorina tangaroa sp. nov.

Figures 1–3, Table 1

Figure 1 Drawings of body regions Chromadorina tangaroa sp. nov.

Chromadorina tangaroa sp. nov. (A) Female anterior body region. (B) and (C) Male cephalic region. (D) Female cephalic region. (E) Female posterior body region. (F) Male posterior body region. Scale bar: 35 μm (A), 16 μm (B–D), 30 μm (E) and 25 μm (F).

Figure 2 Drawings of entire male and female Chromadorina tangaroa sp. nov.

Chromadorina tangaroa sp. nov. (A) Entire female. (B) Entire male. Scale bar: 125 μm (A and B).

Figure 3 Light micrograph of Chromadorina tangaroa sp. nov.

Chromadorina tangaroa sp. nov. Light micrograph of female showing vulva (V), prevulvar pad (PVP) and posterior ovary (PO). Scale bar: 10 μm.

Table 1 Morphometrics of Chromadorina tangaroa sp. nov.

	Males	Females	
	Holotype	Paratype 1	Paratype 2	Paratype 3	Paratype 1	Paratype 2	Paratype 3	Paratype 4	
L	991	763	1,002	847	1,086	910	938	917	
a	29	24	25	26	23	24	24	20	
b	8	6	8	7	8	7	7	7	
c	9	8	9	8	8	7	7	8	
c′	3.4	3.6	3.7	3.5	5.0	5.1	5.1	4.5	
Head diam. at cephalic setae	12	12	14	13	13	14	15	14	
Head diam. at amphids	12	NO	NO	4	16	NO	3	15	
Length of sub-cephalic setae	2–5	4–5	2–3	2–5	4–5	3–5	3–6	3–5	
Length of cephalic setae	6	5–6	6	5	5	5	5	5	
Amphid height	2	NO	NO	2	3	NO	2	2	
Amphid width	4	NO	NO	4	4	NO	3	5	
Amphid width/cbd (%)	33	NO	NO	31	25	NO	20	33	
Amphid from anterior end	1	NO	NO	2	5	NO	3	2	
Nerve ring from anterior end	70	72	70	68	77	74	73	69	
Nerve ring cbd	27	26	27	27	31	28	27	29	
Pharynx length	128	121	128	121	135	131	132	125	
Pharyngeal bulb diam.	28	25	28	26	30	27	27	28	
Pharynx cbd at base	31	29	33	31	36	33	32	33	
Max. body diam.	34	32	40	33	47	38	39	45	
Spicule length	36	32	36	32	–	–	–	–	
Gubernaculum length	35	32	35	32	–	–	–	–	
Cloacal/anal body diam.	33	28	33	30	29	24	25	27	
Tail length	111	101	111	105	144	123	127	122	
V	–	–	–	–	495	427	454	419	
%V	–	–	–	–	46	47	48	46	
Vulval body diam.	–	–	–	–	41	38	39	44	
Note:

Morphometrics (in microns, mean (range)) of Chromadorina tangaroa sp. nov. a, body length/maximum body diameter; b, body length/pharynx length; c, body length/tail length; c′, tail length/anal or cloacal body diameter; cbd, corresponding body diameter; L, total body length; n, number of specimens; NO, not observed; V, vulva distance from anterior end of body; %V, V/total body length × 100.

urn:lsid:zoobank.org:act:5E11A50E-6120-42E4-9836-F2C11D7A02A0

Type locality: Hull of RV Tangaroa (stern), which at the time of sampling was berthed at Burnham wharf in Wellington Harbour, North Island of New Zealand (41.3135 °S, 174.8106 °E).

Type material: Holotype male (NIWA 139243), three paratype males (NIWA 139244, NNCNZ 3331) and four paratype females (NIWA 139244, NNCNZ 3332), collected on 10 November 2019.

Measurements: See Table 1 for detailed measurements.

Description: Male. Body colourless in glycerin preparations, cylindrical, tapering slightly towards anterior extremity. Homogeneously punctated cuticle, with transverse striations approximately 1 μm apart and interspersed with transverse rows of punctations; lateral differentiation absent. Short somatic setae, 3 μm long, sparsely distributed throughout body. Cephalic region blunt, slightly rounded, with relatively well-developed lip region. Inner and outer labial sensilla inconspicuous. Four cephalic setae, 0.3–0.5 cbd long. One or two pairs of sublateral cervical setae present on each side of body, approximately 1.5 cephalic body diameters from anterior extremity. Pigment spots not observed. Amphideal fovea spiral, with 1.5 turns and transversely oval outline, at level of cephalic setae, sometimes difficult to distinguish. Buccal cavity with funnel-shaped pharyngostome; one solid dorsal tooth and two solid, slightly smaller ventrosublateral teeth. Pharynx muscular, not swollen anteriorly, lumen not cuticularized or only slightly cuticularized; conspicuous oval-shape posterior bulb with plasmatic interruptions. Nerve ring located at 55–60% of pharynx length from anterior. Cardia small, surrounded by intestine. Secretory-excretory system with elongated renette cell, 43–69 × 8–14 μm, and small accessory cell, 12–19 × 7–11 μm, both located well posterior to pharynx; pore located far anteriorly at or near level of teeth.

Reproductive system with single anterior outstretched testis located to the right of intestine. Mature sperm cells globular, 9–11 × 6–10 μm, with granular nuclei. Spicules paired, equal, slightly ventrally curved, capitulum rounded, narrow shaft and blade gradually tapering distally; gubernaculum slightly ventrally curved, about as long as spicules, with relatively wide, rounded proximal portion, narrow middle portion, and tapering distal portion. Three small, cup-shaped precloacal supplements present, beginning 5–8 μm anterior to cloacal opening and situated 8–13 μm apart. Precloacal seta not observed. Tail conical, curved ventrally, with few sparsely distributed sublateral setae, 3 μm long; three large caudal glands and well-developed spinneret.

Females. Similar to males, but with slightly lower values of “a” and longer tail. Reproductive system with two opposed, reflexed ovaries; anterior ovary situated to the right of intestine and posterior ovary situated to the left of intestine. Mature eggs 48–51 × 24–32 μm. Vulva not cuticularized, situated slightly anterior to mid-body, not at right angle with body surface but pointing posteriorly; constrictor muscle present. Vagina anteriorly directed. Prevulvar pad present, consisting of area of slightly thicker cuticle with coarse striations located on slightly to conspicuously raised ventral region immediately anterior to vulva. Vaginal glands not observed.

Diagnosis: Chromadorina tangaroa sp. nov. is characterized by body length 763–1,086 μm, cephalic setae 0.3–0.5 cbd long, spiral amphid with 1.5 turns and 20–33% cbd wide, pore of secretory-excretory system located far anteriorly at or near level of teeth. Males spicules with rounded capitulum followed by a narrower shaft and blade tapered distally, gubernaculum as long as spicules, with wide, rounded proximal portion, narrow middle portion, and tapering distal portion, and three cup-shaped precloacal supplements. Females with prevulvar pad, vagina located at 46–48% of body length from anterior, and vagina anteriorly directed.

Differential diagnosis: In addition to the new species, there are four Chromadorina species which possess two or three precloacal supplements: C. obtusa Filipjev, 1918, C. paradoxa Timm, 1961, C. demani Inglis, 1962, and C. micoletzkyi Inglis, 1962. The new species can be differentiated from C. obtusa, C. paradoxa, and C. micoletzkyi by the position of the secretory-excretory pore, which is located well posterior to the buccal cavity but anteriorly to the nerve ring in C. paradoxa and C. micoletzkyi, and posterior to the nerve ring in C. obtusa (vs at level of teeth in C. tangaroa sp. nov.). The new species can also be differentiated from C. paradoxa by the shorter body length (0.76–1.09 vs 1.3 mm), lower ratio of “a” (20–29 vs 37 in C. paradoxa), structure of the posterior pharyngeal bulb (simple vs double in C. paradoxa), longer spicules (32–39 vs 26 μm in C. paradoxa), gubernaculum shape (rounded vs tapering proximal portion in C. paradoxa), and shorter male tail (3.4–3.7 vs 6.7 cloacal body diameters in C. paradoxa), from C. micoletzkyi by the longer body length (0.76–1.09 vs 0.57–0.66 mm in C. micoletzkyi) and higher ratio of “b” (6–8 vs <6 in C. micoletzkyi), and from C. obtusa by the longer spicules (32–39 vs 30 μm), higher ratio of “a” (20–29 vs 17–19 in C. obtusa), and number of precloacal supplements (three vs two supplements in C. obtusa). The position of the secretory-excretory pore in C. demani is not known, but the new species differs from the latter by the longer body length (0.76–1.10 vs 0.63-0.72 mm in C. demani), shorter cephalic setae (0.3–0.5 vs 0.7 cbd in C. demani), relative size of the ventrosulateral teeth (almost the same size as dorsal tooth vs conspicuously smaller than dorsal tooth in C. demani), position of the vulva (46–48 vs 51–55% in C. demani), and shape of the spicules (elongated and straight capitulum vs short and swollen capitulum in C. demani). Chromadorina tangaroa sp. nov. is the first species of the genus to possess a prevulvar pad, although it is conceivable that this feature may have been missed in previous species descriptions.

Etymology: The species is named after RV Tangaroa, on the hull of which the type specimens were found.

Remarks: Specimens were found amongst filamentous brown (Ectocarpales) and green (Chaetomorpha) macroalgae during the survey of RV Tangaroa hull on 10 November 2019. The following invertebrate taxa were also encountered: solitary tunicates, tubeworms, hydroids (Ectopleura and Obelia spp.), the native barnacle Austrominius modestus, the native bivalves Perna canaliculus and Mytilus galloprovincialis, and four bryozoans—the non-indigenous species Cryptosula pallasiana and Watersipora subatra and two species of uncertain identity, viz. Electa oligopora and Celleporaria sp.

Discussion

Chromadorina tangaroa sp. nov. is the first species of the genus to be described from New Zealand. Although the specimens were originally collected from Wellington Harbour, the population found on the hull of RV Tangaroa may have originated from another location in New Zealand, or even overseas. The vessel’s movement in the 4 months prior to sampling include voyages to Hikurangi Margin off the east coast of New Zealand’s North Island, off Campbell Island in the Southern Ocean, and included a 3-day long anchorage in Dunedin on the southeast coast of New Zealand’s South Island. However, during this 4-month period the vessel spent over 4 weeks anchored in Wellington Harbour, which makes it the most likely origin for the hull population.

It seems likely that nematode taxa such as Chromadorina which preferentially occur on macroalgal substrates are transported between ports by shipping. It has been suggested that some cosmopolitan nematode species have been transported outside their native range following the accidental introduction of their macroalgal habitat in new environments (Kim et al., 2019). To date, the genera Graphonema Cobb, 1898, Prochromadora Filipjev, 1922 (family Chromadoridae Filipjev, 1917), and Halomonhystera Andrássy, 2006 (family Monhysteridae De Man, 1876) are the only other nematode genera (beside Chromadorina) that have so far been identified from ship biofouling assemblages (Chan, MacIsaac & Bailey, 2016). Like Chromadorina, these taxa are often associated with attached and drifting macroalgae or artificial substrates (Derycke et al., 2007; Perez-Garcia, Ruiz-Abierno & Armenteros, 2015; Kim et al., 2019). Transport by ballast slurries may also occur; the limited data available to date suggest that they contain nematode assemblages more typical of sedimentary environments with genera such as Desmodora De Man, 1889, Sphaerolaimus Bastian, 1865, and Leptolaimus De Man, 1876 having so far been identified (Radziejewska, Gruska & Rokicka-Praxmajer, 2006).

The present study provides the second record of Chromadorina on the hull of a vessel. Chan, MacIsaac & Bailey (2016) demonstrated that Chromadorina erythrophthalma populations on the hull of military ships can survive long distance transits (1,000s of kilometres covering over 10 degrees of latitude) from Halifax (eastern Canada) to the Arctic (Nanisivik). A study using high-throughput sequencing methods identified the presence of Chromadorina on settlement plates deployed in a New Zealand marina (Von Ammon et al., 2018). Another study showed an unidentified Chromadorina species to be an early colonizer of artificial settlement plates off the coast of Brazil (Fonseca-Genevois et al., 2006). Experiments have shown that Chromadorita tenuis (Schneider, 1906) Filipjev, 1922 (family Chromadoridae) shows a marked preference for macroalgae over sediments, and actively swims several centimetres to colonise macroalgal substrates in response to chemical cues (Jensen, 1981). It appears likely that similar habitat preference and behavior are present in Chromadorina species, but no evidence is yet available to test this hypothesis.

Conclusions

Chromadorina tangaroa sp. nov. is the first species of the genus to be described or recorded from New Zealand, but it is unclear whether it is native to the region because it may have dispersed as part of ship hull biofouling communities. Overall, the potential for shipping to act as dispersal vector for nematodes across ports and oceans remains largely unstudied. The accidental introduction of nematodes to new environments may be a relatively common occurrence, which could explain the cosmopolitan distribution of some species. Nematodes need to be included in studies of ship hull biofouling communities and in biosecurity surveys to gain further insights into the potential effect of shipping on the distribution of nematode species. Although environmental DNA metabarcoding offers a very useful tool for the identification of nematodes from hull samples, incomplete molecular reference datasets limit the ability to identify species or even genera (Holovachov et al., 2017). Furthermore, additional molecular methods will be required to accurately identify describe patterns of genetic connectivity (Darling, Herborg & Davidson, 2012).

I thank Roberta D’Archino, Kate Neill, and Wendy Nelson for providing the nematode specimens, Sarah Allen for survey information, and Dennis Gordon for providing identification of hull fauna. I am grateful to Anna Demchy for obtaining primary literature, to Vadim Mokievsky for translating Russian text, and the three reviewers for providing constructive criticisms on the manuscript.

Abbreviations

a body length/maximum body diameter

b body length/pharynx length

c body length/tail length

c′ tail length/anal or cloacal body diameter

cbd corresponding body diameter

L total body length

n number of specimens

V vulva distance from anterior end of body

%V V/total body length × 100

Additional Information and Declarations

Competing Interests

Author Contributions

Data Availability

New Species Registration

The author declares that they have no competing interests.

Daniel Leduc conceived and designed the experiments, performed the experiments, analyzed the data, prepared figures and/or tables, authored or reviewed drafts of the paper, and approved the final draft.

The following information was supplied regarding data availability:

All of the raw data are available in Table 1 and the figures.

The following information was supplied regarding the registration of a newly described species:

Publication LSID: urn:lsid:zoobank.org:pub:9853D773-4836-43E5-A91D-9AF44C11659D.

Chromadorina tangaroa Leduc.

LSID urn:lsid:zoobank.org:act:46248D45-0D52-4F04-93A0-2B58AF22BCC6.

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
