# Peer review of "A new nematode species, Chromadorina tangaroa sp. nov. (Chromadorida: Chromadoridae) from the hull of a research vessel, New Zealand"

_PeerJ, doi:10.7717/peerj.9233_

## Round 0.1 · original submission · Minor Revisions

The referees only have suggestions for minor revisions. thank you for such a well-written and prepared paper. We look forward to getting your carefully revised MS and responses to the referees' suggestions.

·

Basic reporting

The manuscript contains new and original information: description a new species of free-living nematode from the turf algae on the ship’s hull. The species is new for science and properly described. Apart pure taxonomic interest the manuscript contains important ecological information, which drives attention to the role of meiofauna in fouling communities and to the question of invasive species in meiobenthos and how to recognize them.

Experimental design

Standard methods of material collection were used.

Validity of the findings

The paper is well written and contains all the necessary information. I have just minor comments and corrections.

Additional comments

Lines 57-58: C. epidemos not C. berzicki was reported from macroalgae by Hopper and Meyers, 1967. The last species is from periphyton of Danube River.

The author wrote that genus Cromadorina is “largely marine”. It is correct formally, but the presents of brackish and freshwater species is not mentioned in the text. That drives to confusion when reading lines 51-67. It is not clear, which species in the list are freshwater or brackish.

Line 101: Subfamily name is missed.
Line 157 (and several lines in “Diff. diagnosis”): it is better to put quotation on “a” and other indices.
Line 219: How can the nematode lives on microalgae? Do you mean here the specific life-form of diatoms (turf-like colonies) or else?
Lines 201-206 (or 70-73): Please, provide more taxonomic information about the algae that perform the habitat of the described species, if available, or give a description of the live-form (shape, length of filaments), if not. How this substrate looks like?

Figures are of good quality, and, I guess, the photograph of the head end (and probably, male anal region) will be appropriate here.

·

Basic reporting

no comments

Experimental design

no comment

Validity of the findings

no comment

Additional comments

Main remark: when describing a new species, the holotype should be illustrated. No holotype nor paratypes are indicated in the legend of the figures. Minor corrections (e.g. on terminology, spicule description) are indicated directly on the manuscript.

Reviewer 3 ·

Basic reporting

The description of a new nematode species is basically well done following nematological requriements; the distributional information is of general interest.

The manuscript widely meets all the parts relevant for the basic recording though I have some remarks see below. The manuscript is concisely written for most parts, the literature used is of relevance and mostly properly cited (some references used in the text are missing in the reference list - please check lines 44, 76, 117 and 175); the figures and data given are appropriate.

Adressing the evaluation of the self-containment with relevant results to hypotheses - due to the bipartite character of the paper and basically, with the new description of a species - not a point that can and must be adressed in the manuscript, to my opinion.

Experimental design

Experimental design
Since this is a new description of a species – there is no direct research question, no research gaps other than that this species is new and thus has to be described to fill a so far unknown knowledge gap.
As already mentioned, the data are sufficient for the species new description and follow common standards of nematode species description.
Methods – sorting, slide preparation and measurements (inclusively magnification) could be given in more detail to my opinion. How have nematodes been separated from the macrophytes – sieving, centrifugation, have samples been stained, etc?
The biogeography part lacks investigation – as already mentioned, this is one difficult to handle point of the manuscript – since the discussion is about the dispersal and distribution, but the species dispersal is nothing which has been investigated in the study.

Validity of the findings

The new species descriptions based on morphological data alone seems to be justified with regard to the differentiation of four other species with similar male supplements.
As also mentioned, conclusion is difficult to handle since there is no original research question other than the species description – deals with the biogeography part of the paper.
And this final comment address once again my major point of concern – the bipartite paper – mixing a species new description with some biogeographical aspects. Maybe choice of another section heading (biogeography aspects) could circumvent this issue if this finds editor’s (journal) approval?

Additional comments

Some more basic remarks on the manuscript - Difficulties arose for me as a reviewer confronted with the two differnt contents of the manuscript - the species description and the (potential) dispersal, where yet the latter has not actually been investigated. To my opinion, in order to combine these two different contents in one manuscript, an approach might help to overcome this issue, namely, slightly deviating naming of sections – e.g. instead of results, which are generally lacking in this paper, taxonomy or species description as one section title, and instead of discussion – ecology or biogeography – but only if this would meet editor’s or journal’s agreement.

This “two parted content” causes my only one major concern for accepting publication after minor revision; since the distribution has actually not been investigated and is more a side effect discussion of the taxa adressed within the species description. Nevertheless, both parts are of relevance and informative for nematode taxonomy and biogeography.


Additional remarks:

Please correct:
author of order - line 99,
subfamily name - line 101,
a instead of A for Cobb’s measures line 79.

Suggestions for the introduction: I would recommend to distinguish between biofilm and biofouling – what are the differences? Are biofilms or macroalgae “aufwuchs” automatically biofouling substrates? One to two sentences on this topic would be helpful to my opinion – e.g. “Introduction” starts with 32 with biofouling though “this biofouling communities” are not well introduced and explained; biofouling to my understanding is something which degrades “technical materials” – is this given for the macrophytes and their inhabitants with regard to ship vessels? Please clarify this passage on the biofouling” vs. biofilm or macrophyte dwellers.

I would also suggest to position “Etymology” – before “type locality” at the beginning of the species description (e.g. line 122).

Wording: though it is commonly used I would recommend to clarify what a homogenous cuticle is in the context of the Chromadorida taxa – namely a homogenous punctuation or homogenously punctated cuticle (line 104, line 135). Please avoid homogenous cuticle.

Discussion:
e.g. omit 218-225 somehow redundant - see introduction; not essential information to my opinion

Rearrangement of paragraph from 239 consecutive to line 217 – might fit better considering readability.

One question came to my mind when reading the manuscript – what do we know of other marine Chromadorina species recorded from New Zealand, are the common, widely distributed, restricted to some specific areas, can their distribution give indication / speculation of narrow ranges etc.
It would be of interest to have some additional information on this, if available, somewhere within the ecological part.

---

## Round 0.2 · accepted · Accept

Thank you for submitting a high-quality paper and promptly addressing the minor suggestions of the referees. thank you for choosing PeerJ.